# Bridge Nodes between Personality Traits and Alcohol-Use Disorder Criteria: The Relevance of Externalizing Traits of Risk Taking, Callousness, and Irresponsibility

**DOI:** 10.3390/jcm11123468

**Published:** 2022-06-16

**Authors:** Ana De la Rosa-Cáceres, Marta Narvaez-Camargo, Andrea Blanc-Molina, Nehemías Romero-Pérez, Daniel Dacosta-Sánchez, Bella María González-Ponce, Alberto Parrado-González, Lidia Torres-Rosado, Cinta Mancheño-Velasco, Óscar Martín Lozano-Rojas

**Affiliations:** 1Departamento de Psicología Clínica y Experimental, University of Huelva, 21071 Huelva, Spain; ana.delarosa@dpces.uhu.es (A.D.l.R.-C.); marta.narvaez885@alu.uhu.es (M.N.-C.); andrea.blanc@dpces.uhu.es (A.B.-M.); nehemias.romero@dpces.uhu.es (N.R.-P.); daniel.daco@dpces.uhu.es (D.D.-S.); bellamaria.gonzalez@dpces.uhu.es (B.M.G.-P.); alberto.parrado@dpces.uhu.es (A.P.-G.); lidia.torres@dpces.uhu.es (L.T.-R.); cinta.mancheno@dpces.uhu.es (C.M.-V.); 2Research Center for Natural Resources, Health and Environment, University of Huelva, 21071 Huelva, Spain

**Keywords:** alcohol-use disorders, personality disorders, externalizing, network analysis, antisocial personality disorder, borderline personality disorder

## Abstract

Background: Personality disorders show strong comorbidities with alcohol-use disorder (AUD), and several personality traits have been found to be more frequent in people with AUD. This study analyzes which personality facets of those proposed in the Alternative Model of Personality Disorder (AMPD) of DSM-5 are associated with the diagnostic criteria of AUD. Methods: The sample was composed of 742 participants randomly selected from the Spanish population, and 243 patients attending mental health services. All participants were of legal age and signed an informed consent form. The instruments were administered to the community sample in an online format, and a psychologist conducted individual face-to-face interviews with the patients. AMPD facets were assessed through the Personality Inventory of DSM-5 Short-Form, and the AUD criteria through the Substance Dependence Severity Scale. A network analysis was applied to identify the personality facets mostly associated with the AUD criteria. Results: The network analysis showed the existence of three communities, grouping the AUD criteria, externalizing spectrum facets, and internalizing spectrum facets, respectively. Risk taking, callousness, and irresponsibility facets showed the strongest association with the AUD criteria, bridging externalizing personality traits with AUD criteria. Conclusions: The facets of risk taking, callousness, and irresponsibility should be accurately assessed in patients with AUD to differentiate between a possible primary personality disorder and a syndrome induced by alcohol addiction.

## 1. Introduction

Alcohol is the most widely consumed drug worldwide [1], yet only a fraction of alcohol consumers develop an alcohol-use disorder (AUD). Various factors have been suggested to increase the likelihood of developing AUDs, including genetic [2], social [3], neuropsychological [4], and psychopathological and personality traits [5].

Focusing on the latter, population-based studies show that comorbidity between AUD and PDs exceeds 40% [6], with the highest prevalence rates detected with antisocial, histrionic, and borderline personality disorders (PDs). Studies conducted with patient samples show that the prevalence rates of comorbid disorders are higher than those observed in the general population, noting that greater severity of AUD is associated with a higher probability of presenting these disorders [7]. A review conducted by Guy et al. [8] showed that in patients diagnosed with antisocial PD, lifetime AUD reached 76.7%, while among patients diagnosed with borderline PD the prevalence of AUD was 52.2%. In patients with AUD, studies have reported mixed prevalence rates. However, Trull et al. [9] estimate that in patients diagnosed with AUD, the general prevalence of PD exceeds 45%. Likewise, these authors point out that in AUD patients, Cluster B PD is more prevalent than clusters A and C.

The above evidence indicates that several personality traits are likely to be shared among individuals with AUD and PD. In this regard, numerous studies have been conducted using the Five-Factor Model (FFM) to determine which traits are characteristic of heavy alcohol users and those with AUD. For instance, a meta-analysis conducted by Malouff et al. [10] showed that low conscientiousness, low agreeableness, and high neuroticism were associated with alcohol consumption. A subsequent meta-analysis by Kotov et al. [11] also found that low conscientiousness and high neuroticism traits were more frequently found in people with AUD. However, no association was found with the agreeableness trait. Moreover, the meta-analysis by Hakulinen et al. [12] also agrees with previous studies by showing that lower agreeableness and conscientiousness and higher neuroticism are associated with heavy alcohol consumption. In addition, these authors also found that higher extraversion is associated with heavy alcohol consumption in a specific way; that is, the traits associated with the transition from moderate to heavy alcohol consumption were lower conscientiousness and higher extraversion. The latter trait, aligned with the detachment domain of the Alternative Model for Personality Disorders (AMPD), was found by Moraleda et al. [13] to be a distinctive trait of patients with AUD compared to those with other substance-use disorders. However, the described relationships between personality traits and AUD should be contextualized in those countries where alcohol consumption is widely accepted in the culture. Factors associated with disapproval of alcohol consumption, differences in family and interpersonal values, or attitudinal aspects that differ across cultures may mediate the relationships between personality traits and alcohol consumption [14].

Despite previous evidence suggesting that people with high alcohol consumption exhibit certain personality traits, the specific relationships between these traits and AUD are largely unknown. In this regard, network analysis could help to delve deeper into the relationships between personality traits and AUD. Although this type of technique has its origins in sociological studies, in the last decade it has been applied to the study of mental disorders [15]. Network analysis constitutes a set of techniques that allow the reciprocal relationships between symptoms or diagnostic criteria to be depicted in graphical form. Each symptom or diagnostic criterion is represented by a node, allowing for analysis of the interrelationships between these nodes [16,17]. Those nodes that are more densely related form substructures or clusters [18], which can be distinguished from other possible clusters. In addition, this technique allows us to determine which criteria or symptoms exert a greater influence on the others [19] and identify those nodes that are most strongly related to nodes of other distinct substructures. These nodes are considered useful for explaining comorbidity between various disorders, as shown by previous studies that have used network analysis to depict associations between AUD and internalizing traits [20].

Thus, the present study aimed to (1) examine the relationships between personality facets and AUD criteria according to their organization into different substructures in the network, as well as to test for invariance according to gender by comparing the structure, global strength, and edges between the networks of men and women; and (2) identify the bridge nodes between the different substructures identified, which could help to explain the comorbidity between PD and AUD.

## 2. Methods

### 2.1. Participants

The present study sample included 985 participants, composed of adults randomly selected from the general population (*n* = 742) and a sample of patients undergoing treatment at mental health services (*n* = 243).

The 742 adults in the community sample were selected by stratified random sampling, proportionally represented in the Spanish population according to age group, gender, and geographic area. The inclusion criteria for the community sample were being over 18 years old and not presenting any diagnosis of mental disorder. The 243 patients in the sample were being treated in public mental health services in the province of Huelva (Spain). The inclusion criteria for the patient sample were being over 18 years old and undergoing treatment in the mental health services during data collection. The exclusion criteria for both samples (patients and community) were (1) having been diagnosed with a medical or psychological disorder that disqualified them from taking the tests; and (2) not signing the informed consent form.

Sociodemographic characteristics of the total sample are presented in Table 1. Of the sample, 49.7% (*n* = 490) were female, with an age range of 18 to 80 years (*M* = 44.93; *SD* = 14.6). On the other hand, 24.26% of the sample met the diagnostic criteria for at least one mental disorder according to the Diagnostic and Statistical Manual of Mental Disorders, fifth edition (DSM-5). Table 2 shows the frequency and proportion of diagnoses for the patient sample (*n* = 243) and the whole sample (*n* = 985), with depressive disorders (38.68%) and anxiety disorders (36.21%) being the most frequent.

### 2.2. Measures

*The Personality Inventory for DSM-5-Short Form* -PID-5-SF- [21] in its Spanish version [22]. The PID-5-SF assesses the 25 personality facets identified in the Alternative Model of Personality Disorder (AMPD) of the DSM-5: Anhedonia, Anxiousness, Attention Seeking, Callousness, Deceitfulness, Depressivity, Distractibility, Eccentricity, Emotional Lability, Grandiosity, Hostility, Impulsivity, Intimacy Avoidance, Irresponsibility, Manipulativeness, Perceptual Dysregulation, Perseveration, Restricted Affectivity, Rigid Perfectionism, Risk Taking, Separation Insecurity, Submissiveness, Suspiciousness, Unusual Beliefs and Experiences, and Withdrawal. These facets are organized into five higher-order domains: Negative Affect, Detachment, Antagonism, Disinhibition, and Psychoticism. The 25 facets are assessed through 100 Likert-response format items (four items per facet) from 0 (“*very false or often false*”) to 3 (“*very true or often true*”). Higher scores indicate a greater presence of the facets.

This instrument has shown adequate test–retest reliability and internal consistency. Likewise, according to the Standards for Educational and Psychological Testing [23], evidence has been provided on its internal structure and relationship with other variables [22].

For the sample used in this study, Cronbach’s Alpha coefficient values above 80 were found for 14 of the 25 facets; another nine facets showed internal consistency values above 0.70, and only two facets presented internal consistency values below this value (Callousness: α = 0.69 and Irresponsibility: α = 0.63).

Spanish version of the *Substance Dependence Severity Scale -SDSS- for DSM-5* [24,25]: The SDSS consists of a semistructured interview designed to assess the severity of dependence on one or more substances [26]. This instrument evaluates the diagnostic criteria established in the DSM-5, using an evaluation timeframe of 30 days prior to the interview.

The Spanish version of the SDSS has shown evidence of good psychometric properties in terms of reliability and validity [24,25]. In this study, only the items that operationalize the 11 diagnostic criteria of the AUD were administered, which were coded with the values 1 (presence) and 0 (absence). A reliability value (estimated through Cronbach’s alpha) of 0.93 was obtained for the study sample.

In addition, questions were included on sociodemographic variables related to gender, age, educational level, and employment.

### 2.3. Procedure

The instruments were administered to the community sample in an online format. Before administration, participants were asked to answer a series of questions to check that they could read and write correctly.

A trained psychologist administered the instruments to the sample of patients through individual interviews that took place in a room in the mental health center where they were being treated.

All participants (community sample and patients) were informed of the study objectives and the voluntary and anonymous nature of their participation and signed the informed consent form before completing the instruments. This study has been approved by the Bioethics Committee of Biomedical Research of Andalusia (Spain) (file number PI 040/18).

### 2.4. Data Analysis

The multivariate normality test revealed the absence of multivariate normality for skewness (Mardia = 60,874.19) and kurtosis (Mardia = 272.67). Therefore, the network was estimated using the GLASSO algorithm [27] in combination with the EBIC selection model (hyperparameter γ = 0.5) [28] applied to the nonparanormal transformation of the data set [29]. For the layout of the graph, the Fruchterman–Reingold algorithm was used [30]. The study’s sample size (*n* = 985) was adequate to estimate the network according to the simulation study [31] (Appendix A).

To detect community structures, the walktrap algorithm was employed [32]. The strength of centrality indices, one-step Expected Influence (EI1), and two-step Expected Influence (EI2) were estimated [33]. Strength and EI provide information on the direct relationships between each node and the rest by summing the weights of the edges, considering the absolute values or the sign of the value (positive or negative), respectively. EI2 also sums the weights of indirectly related edges [33]. The stability of the centrality indices was quantified using a person-dropping bootstrap procedure that provides a correlation-stability coefficient (CS-coefficient). CS-coefficient values > 0.5 indicate strong stability and interpretability [31].

In addition, predictability (i.e., the proportion of variance of each node that is explained by its neighboring nodes) was estimated [34], along with the Participation Coefficient (PC), and Participation Ratio (PR) [35]. Higher PC values indicate that the edges of the nodes are distributed more equally among the network communities, while higher PR values indicate nodes with more numerous and stronger edges. The PC and PR values were transformed to a scale of 0 to 1 for ease of interpretation.

Regarding the bridge nodes, the bridge strength, bridge EI1, and bridge EI2 were estimated [36]. Bridge strength and bridge EI1 indicate the total connectivity of each node with nodes of other communities with which it is directly related, by summing the weights of the edges that connect the node with nodes of other communities considering absolute values (bridge strength) or the sign of the values (bridge EI1). Bridge EI2 also considers indirect relationships with nodes in other communities. A blind cut-off point at the 80th percentile of bridge strength was applied to identify bridge nodes [37].

Finally, network invariance for men (*n* = 495) and women (*n* = 490) was analyzed using the network comparison test [38] (5000 times repeated subsampling). In addition, the invariance of network structure and overall strength was analyzed. The M statistic indicates the differences in the connections between the edges of both networks, while the S statistic indicates the difference in global strength between the two networks. If the test for network-structure invariance is significant, the invariance of the individual edges will be examined [38].

All analyses were conducted in R 4.1.2 and R-Studio 2022.2.0.443. In addition, the packages mvn 5.9 [39], bootnet 1.5 [31], igraph 1.3.0 [40], network tools 1.4.0 [36], qgraph 1.9.2 [41], and Network-ComparisonTest 2.2.1 [38] were used.

## 3. Results

### 3.1. Network Structure with AMPD Personality Facets and AUD Criteria, and Invariance According to Gender

The estimated network is shown in Figure 1. The network consisted of 250 edges (out of a possible 630) that showed a partial correlation value different from zero. The weights of the edges ranged from −0.009 (Restricted Affectivity-Attention Seeking) to 0.431 (Depressivity-Anhedonia). The graphical analysis reveals an optimal 3-community solution (modularity index = 0.45), which organizes the nodes according to a structure that differentiates the facets most strongly linked to the internalizing spectrum (Anhedonia, Anxiousness, Depressivity, Distractibility, Eccentricity, Emotional Lability, Hostility, Intimacy Avoidance, Perceptual Dysregulation, Perseveration, Restricted Affectivity, Rigid Perfectionism, Separation Insecurity, Submissiveness, Suspiciousness, Unusual Beliefs and Experiences, and Withdrawal), to the externalizing spectrum (Attention Seeking, Callousness, Deceitfulness, Grandiosity, Impulsivity, Irresponsibility, Manipulativeness, and Risk Taking), and the AUD diagnostic criteria (Quit/Control, Time Spent, Activities Given Up, Tolerance, Withdrawal, Larger/Longer, Physical/Psychological Problems, Neglect Roles, Hazardous Use, Social/Interpersonal Problems, and Craving).

The standardized scores of the centrality indices, bridge-centrality indices, and the predictability of the nodes are displayed in Table 3. The nodes with the highest strength values were Anxiousness (1.94), Anhedonia (1.67), Perseveration (1.51), and Emotional lability (1.08). The highest EI1 values were for Perseveration (1.51), Anhedonia (1.48), Anxiousness (1.43), and Physical/ Psychological Problems (1.23). The highest EI2 values corresponded to Perseveration (1.57), Anhedonia (1.50), Eccentricity (1.24), and Anxiousness (1.20). The CS-coefficient (cor = 0.7) was 0.75 for Strength and 0.75 for EI (see Appendix A, respectively), indicating strong stability and interpretability of the estimates. The nodes with the highest bridge strength, bridge EI1, and bridge EI2 values were Irresponsibility (bridge strength = 2.77; bridge EI1 = 2.91; bridge EI2 = 3.01), Impulsivity (2.49, 2.85, 2.84), Callousness (2.21, 1.34, 1.25), and Risk Taking (1.43, 1.47, 1.82). The predictability of each node is shown in Figure 1 and Table 3. The predictability values (R^2^) ranged from 0.62 (Anhedonia) to 0.22 (Separation Insecurity), with an average value of 0.42. The symptoms with the least explained variance (i.e., the most independent) were Separation Insecurity (R^2^ = 0.22), Quit/Control (R^2^ = 0.24), Intimacy Avoidance (R^2^ = 0.24), Larger/Longer (R^2^ = 0.27), and Restricted Affectivity (R^2^ = 0.27). In contrast, the symptoms with the greatest explained variance were Anhedonia (R^2^ = 0.62), Depressivity (R^2^ = 0.60), Anxiousness (R^2^ = 0.57), and Perseveration (R^2^ = 0.56). Table 3 also shows the PC and PR values. The highest PC values correspond to Hostility (PC = 0.53), Impulsivity (PC = 0.50), and Rigid Perfectionism (PC = 0.38), these being the nodes with edges that are distributed more equally among the communities. According to the PR values, the nodes with the strongest and most numerous edges were Irresponsibility (PR = 1), Perseveration (PR = 0.97), and Perceptual Dysregulation (PR = 0.90).

Concerning the invariance analysis, Table 4 shows the descriptive statistics (means and standard deviations) for the AMPD facets and AUD criteria in the total sample (*n* = 985), the male subsample (*n* = 495), and the female subsample (*n* = 490). It is observed that the mean difference between males and females has a small or null effect size for all AUD facets and criteria except for Emotional Lability, which yielded a medium effect size (*d* = 0.62). 

The above values are congruent with the invariance test conducted for men (*n* = 495) and women (*n* = 490), detecting no significant differences in terms of network structure (i.e., differences in the edge connections of the two networks, M = 0.22; *p* = 0.838) or global strength (i.e., difference in the sum of the absolute weights between the two networks, S = 0.34; *p =* 0.494). The overall strength of the network estimated for the male sample was 17.18, and for the female sample this was 16.84.

### 3.2. “Bridging Nodes” between AUD Criteria and Personality Facets

Figure 2 shows (in yellow) the facets that act as “bridge nodes” between the different network structures (i.e., the nodes more connected to nodes of other communities with which they are directly related), according to the highest values of bridge strength shown in Table 3. Through the edges, it is observed that most of the relationships between the AUD criteria and the personality facets occur through these “bridge nodes”. Furthermore, it appears that most of the facets acting as “bridge nodes” correspond to the “antagonism” and “disinhibition” domains of the AMPD. Only the bridge node corresponding to the “hostility” facet falls within the “negative affectivity” domain of the AMPD.

A detailed analysis of the relationships between the AUD criteria and the bridging nodes is displayed in Table 5, which shows the partial and Pearson correlations between the facets that constitute “bridging nodes” and the AUD diagnostic criteria. The bridge node “risk-taking” is the one that presents the most relationships with the AUD diagnostic criteria, followed by the facets of “irresponsibility” and “callousness.” Concerning the AUD diagnostic criteria, it can be observed that the diagnostic criterion “withdrawal” shows the most associations with the bridging nodes, together with “neglect roles” and “craving”.

## 4. Discussion

The present study aimed to deepen our knowledge of the existing comorbidity between PDs and AUD. For this purpose, this study aimed to complement the existing evidence [10,11,12] with new evidence obtained through network analysis and use of the DSM-5 AMPD trait model. To our knowledge, no previous studies have used network analysis to examine the relationship between the DSM-5 AMPD traits and AUD criteria.

Analysis of the centrality indices of the AUD criteria has shown relatively low values, indicating their low capacity to influence the personality facets. This finding is also evident in the visualization of the network, in which the AMPD facets and the AUD criteria are organized into three independent communities—albeit empirically related and theoretically grounded. The alcohol diagnostic criteria maintain close relationships, leading them to be organized into an independent community, while the AMPD facets are organized into two interrelated communities, linked to the internalizing and externalizing spectrum. This organization of the AMPD facets is consistent with studies that have applied a hierarchical analysis of personality [42,43,44], and the independent organization of the AUD is congruent with the Hierarchical Taxonomy of Psychopathology (HiTOP) model [45] and previous network analysis studies [46,47,48]. Despite the relative independence of the three communities, the AUD diagnostic criteria show relationships with a set of personality facets that act as a bridge. These are mostly associated with the externalizing spectrum, highlighting the relationships with Callousness, Irresponsibility, and Risk Taking.

Our observation of these bridging facets is congruent with the relationships observed in previous studies analyzing FFM traits in alcohol consumers. The specialized literature supports an association between alcohol consumption and the FFM traits of conscientiousness and agreeableness [10,12], which are aligned with the disinhibition and antagonism domains of the AMPD [49]. This study has shown that these relationships could largely be due to the connection between the AUD criteria and the bridging facets within the disinhibition (risk taking, irresponsibility, impulsivity, and rigid perfectionism) and antagonism (callousness, grandiosity, and hostility) domains. Similarly, it is worth noting the connection between the AUD criterion “withdrawal” and the facet “hostility” framed in the negative affectivity domain of the AMPD (aligned with the neuroticism trait included in the FFM). This finding is congruent with the results of the meta-analysis conducted by Hakulinen et al. [12], which found that the domain “neuroticism” is associated primarily with people who engage in heavy alcohol consumption. It is also compatible with studies suggesting that people with AUD have biases toward recognizing and attributing hostile expressions and behaviors [50,51]. Thus, one hypothesis to explain this relationship could be that the occurrence of “withdrawal” exacerbates the “hostility” trait. Therefore, this association is observed mainly in heavy alcohol users and is weaker in those who consume alcohol in moderation.

On the other hand—and concerning PDs—some studies show that it is borderline and antisocial PDs that most frequently co-occur with AUD. According to the DSM-5 AMPD [52], antisocial PD is evaluated based on the presence of the facets “Callousness”, “Irresponsibility”, and “Risk Taking”, among others, while Borderline PD includes the facets of “Risk Taking” and “Hostility” for its diagnosis. Thus, the bridging facets identified in this study are some of those required for the assessment of the two PDs mentioned above. Therefore, it is likely that the high comorbidity of AUD with these disorders is caused by the relationships between the diagnostic criteria of AUD and these bridging facets. This finding along with the results reported by other authors [53,54], could also help to explain why patients with antisocial and borderline PD have higher rates of relapse and treatment dropout. These authors found that patients who prematurely terminate treatment score higher on the facets of “Hostility”, “Callousness”, and “Risk-taking” than those who complete treatment. Thus, these bridging facets that are relevant to antisocial and borderline PD diagnoses, are also associated with premature patient dropout.

We consider the findings of this study to be useful for advancing our knowledge of the comorbidity of AUD with PDs. Nonetheless, it is also worth considering some aspects of the study sample that could have affected the results. As already indicated, the sample of patients shows the highest prevalence of emotional disorders (depressive and anxiety disorders). This may have maximized the relationships between the facets associated with the “negative affectivity” domain, which resulted in higher values of the centrality indices observed for these facets. However, aside from the centrality indices, the network structure is congruent with the hierarchical analysis reported in previous studies [42,43,44], and so the impact of this sample composition is more likely to be limited to the understanding of comorbidity between AUDs and PDs.

We would also like to point out that the partial correlations between the “bridging nodes” and the AUD criteria are low. It is possible that these weak relationships were found because this study did not include a specific sample of patients with AUDs, with the sample of the general population being more representative. Consequently, greater variability in the personality facets and AUD criteria are observed compared to the case in which AUD patients are specifically selected, which could negatively affect the values of the partial correlations. Despite this, the results have revealed the existence of “bridging nodes” congruent with evidence from previous meta-analysis [10,11,12]. Therefore, we consider that the results of this study provide novel insights that could help to improve our understanding of comorbidity between these disorders. 

Finally, we consider it necessary to limit the generalizability of the relationships observed in this study to the cultural context in which it was carried out. In this sense, it should be noted that cultural studies analyzing the relationship between personality traits and alcohol consumption have shown differences in the relationships established between agreeableness, antisocial behavior, and alcohol-related consequences when comparing different countries [55]. Moreover, it should be considered that social norms about what constitutes, for example, “irresponsibility” or “risk taking” in the case of personality trait assessment, or “social/interpersonal problems” or “hazardous use” in the case of AUD assessment, may differ across cultures and even within a culture [56]. Therefore, we consider that the observed interrelationships be contextualized within the framework of Spanish culture as well as in other countries with equivalent social norms and legislation. Future cross-cultural studies should provide evidence on the stability of these relationships in other countries and cultures.

## 5. Conclusions

The present study has shown, through network analysis, connections between AUD diagnostic criteria and personality facets. It has been observed that the bridge nodes correspond to facets associated with the disinhibition and antagonism personality domains of the AMPD. This finding is congruent with the results of previous studies using the FFM model and applying other statistical techniques. This finding also helps to understand the comorbidities observed between AUD, borderline, and antisocial personality disorder. From a clinical perspective, the results indicate the importance of accurately assessing the risk-taking, callousness, and irresponsibility traits in patients with AUD, in order to differentiate between a possible primary personality disorder, or the existence of a syndrome induced by alcohol addiction. In addition, the observed connections may be useful to guide the development of interventions aimed at dual-pathology patients with AUD and PD.

## Figures and Tables

**Figure 1 jcm-11-03468-f001:**
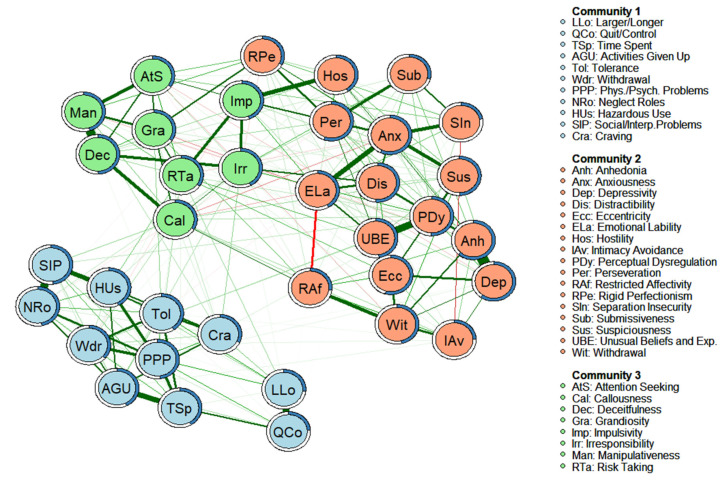
Empirical network model (network structure estimated from a walktrap modularity analysis) for the complete sample (*n* = 985). Note. Each node represents a symptom. The edges represent the relationships (partial correlations) between the symptoms. Positive relationships are represented in green, and negative relationships in red. The thickness of the edge reflects the strength of the association, so that the most strongly related symptoms are connected by thicker edges. The blue pie chart surrounding each node represents the predictability of each node (a higher proportion of blue indicates greater predictability). The membership of the nodes to the different communities is represented by different colors: the symptoms of Community 1 are shown in blue, the symptoms of Community 2 are shown in salmon, and the symptoms of Community 3 are shown in green. The arrangement of the nodes was established based on the Fruchterman-Reingold algorithm.

**Figure 2 jcm-11-03468-f002:**
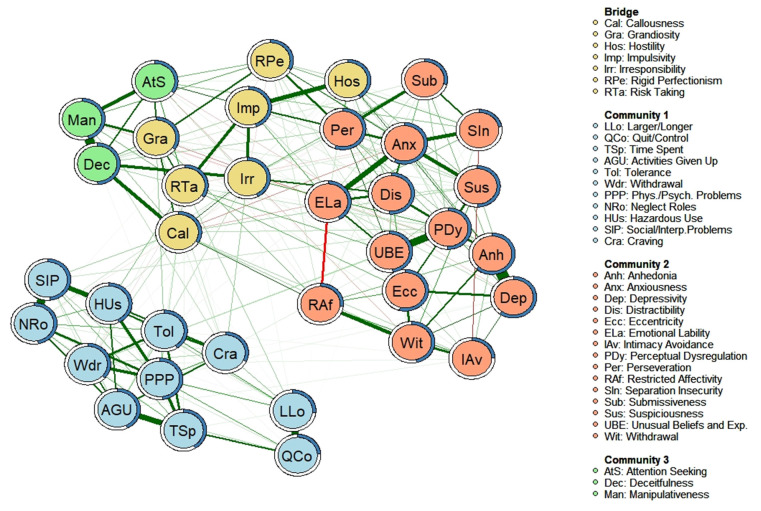
Empirical network model (network structure estimated from a walktrap modularity analysis) and bridge nodes for complete sample (*n* = 985). Note. Each node represents a symptom. The edges represent the relationships (partial correlations) between the symptoms. Positive relationships are represented in green. and negative relationships in red. The thickness of the edge reflects the strength of the association, so the most strongly related symptoms are connected by thicker edges. The blue pie chart surrounding each node represents the predictability of each node (a higher proportion of blue indicates greater predictability). The membership of the nodes to the different communities is represented by different colors: bridge symptoms are shown in yellow; the symptoms of Community 1 are shown in blue; the symptoms of Community 2 are shown in salmon; and the symptoms of Community 3 are shown in green. The arrangement of the nodes was established based on the Fruchterman-Reingold algorithm.

**Table 1 jcm-11-03468-t001:** Sociodemographic characteristics of participants.

Sociodemographic Characteristics	*n*	*%*
Gender		
Male	495	50.3
Female	490	49.7
Education level		
Did not complete primary education	17	1.7
Primary education	52	5.3
Secondary education	533	54.1
University studies	383	38.9
Employment status		
Employed	569	57.8
Unemployed	416	42.2

**Table 2 jcm-11-03468-t002:** Distribution of diagnoses in the patient sample.

	*n*	% Sample Patients (*n* = 243)	% All Sample (*n* = 985)
Neurodevelopmental Disorders	5	2.06	0.51
Schizophrenia Spectrum and Other Psychotic Disorders	21	8.64	2.13
Bipolar and Related Disorders	11	4.53	1.12
Depressive Disorders	94	38.68	9.54
Anxiety Disorders	88	36.21	8.93
Obsessive-Compulsive and Related Disorders	6	2.47	0.61
Trauma- and Stressor-Related Disorders	76	31.28	7.72
Dissociative Disorders	2	0.82	0.20
Somatic Symptom and Related Disorders	2	0.82	0.20
Feeding and Eating Disorders	4	1.65	0.41
Substance-Related and Addictive Disorders	10	4.12	1.02
Personality Disorders	25	10.29	2.54

**Table 3 jcm-11-03468-t003:** Centrality indices and bridge-centrality indices of personality facets and alcohol-use disorder criteria.

Nodes	Strength	Expected Influence 1	Expected Influence 2	Bridge Strength	Bridge Expected Influence 1	Bridge Expected Influence 2	R^2^	PC	PR
**Personality Facets of Alternative Model of Personality Disorder**									
Anhedonia	1.67	1.48	1.50	−1.00	−1.14	−1.02	0.62	0.00	0.61
Anxiousness	1.94	1.43	1.20	−0.82	−1.33	−1.00	0.57	0.01	0.71
Attention Seeking	−0.21	−0.57	−0.64	0.32	−0.39	−0.14	0.34	0.11	0.45
Callousness	0.40	−0.24	−0.32	2.21	1.34	1.25	0.33	0.29	0.86
Deceitfulness	0.83	1.03	0.97	−0.33	−0.09	0.49	0.50	0.02	0.58
Depressivity	0.91	0.82	1.10	−0.89	−1.04	−1.00	0.60	0.00	0.59
Distractibility	0.35	0.61	0.75	−0.18	0.04	−0.12	0.51	0.15	0.64
Eccentricity	1.04	1.20	1.24	−0.46	−0.22	−0.08	0.54	0.05	0.67
Emotional Lability	1.08	−1.46	−1.19	0.20	−1.06	−1.25	0.41	0.09	0.86
Grandiosity	0.12	−0.10	−0.26	1.36	0.95	0.83	0.34	0.25	0.49
Hostility	−0.93	−0.46	−0.26	0.77	1.05	0.63	0.40	0.53	0.51
Impulsivity	0.16	0.47	0.41	2.49	2.85	2.84	0.44	0.50	0.37
Intimacy Avoidance	−1.75	−2.09	−2.01	−0.60	−0.37	−0.53	0.24	0.04	0.28
Irresponsibility	0.63	0.69	0.73	2.77	2.91	3.01	0.42	0.37	1
Manipulativeness	0.39	0.25	0.22	0.02	−0.27	−0.11	0.45	0.04	0.40
Perceptual Dysreg.	0.61	0.84	0.95	0.60	0.88	0.83	0.50	0.09	0.90
Perseveration	1.51	1.51	1.57	0.34	0.49	0.47	0.56	0.15	0.97
Restricted Affectivity	−0.49	−1.75	−1.72	0.29	0.43	0.15	0.27	0.18	0.35
Rigid Perfectionism	−0.75	−0.68	−0.70	0.96	0.75	0.60	0.33	0.38	0.48
Risk Taking	−0.17	−0.02	0.02	1.43	1.47	1.82	0.35	0.23	0.67
Separation Insecurity	−1.58	−2.15	−2.15	−0.53	−0.57	−0.72	0.22	0.05	0.20
Submissiveness	−1.84	−1.28	−1.28	−0.42	−0.28	−0.32	0.29	0.11	0.17
Suspiciousness	0.34	0.61	0.77	−0.40	−0.17	−0.15	0.50	0.04	0.85
Unusual Bel. and Exp.	−0.02	0.31	0.38	−0.05	0.21	0.30	0.48	0.05	0.48
Withdrawal	0.27	0.20	0.06	−0.56	−0.82	−0.74	0.46	0.03	0.39
Alcohol-Use Disorders									
Larger/Longer	−1.54	−0.97	−1.23	−0.78	−0.55	−0.69	0.27	0.02	0.15
Quit/Control	−2.24	−1.59	−1.75	−1.14	−0.97	−1.07	0.24	0.00	0
Time spent	−0.36	0.03	0.01	−1.08	−0.88	−0.88	0.41	0.00	0.31
Activities given up	0.05	0.37	0.23	−0.74	−0.52	−0.53	0.43	0.02	0.36
Tolerance	0.12	0.36	0.22	−0.56	−0.44	−0.43	0.41	0.00	0.61
Withdrawal	−0.61	−0.35	−0.26	−0.15	−0.12	−0.21	0.37	0.03	0.56
Phys./Psych.Problems	1.07	1.23	0.99	−0.79	−0.57	−0.54	0.48	0.00	0.55
Neglect roles	0.06	0.19	0.28	−0.81	−0.85	−0.81	0.47	0.02	0.60
Hazardous use	−0.18	0.18	0.21	−0.56	−0.32	−0.35	0.43	0.02	0.27
Social/Interp.Prob.	0.18	0.48	0.50	−0.66	−0.43	−0.49	0.48	0.01	0.38
Craving	−1.07	−0.57	−0.54	−0.22	0.03	−0.04	0.32	0.04	0.44

Note. R^2^ = Predictability; PC = Participation Coefficient; PR = Participation Ratio.

**Table 4 jcm-11-03468-t004:** Means, standard deviations, and Cohen’s d of scores on the PID-5-SF subscales and alcohol-use disorder criteria.

	All Sample	Men	Women	Cohen’s d
PID-5 Subscales				
Anhedonia	6.78 (3.24)	6.18 (2.71)	7.38 (3.60)	0.38
Anxiousness	8.60 (3.38)	7.81 (3.17)	9.41 (3.39)	0.49
Attention Seeking	5.88 (2.50)	6.04 (2.55)	5.72 (2.44)	0.13
Callousness	4.75 (1.50)	4.90 (1.69)	4.59 (1.26)	0.34
Deceitfulness	5.24 (2.00)	5.40 (2.16)	5.08 (1.81)	0.16
Depressivity	5.88 (2.91)	5.39 (2.36)	6.38 (3.31)	0.34
Distractibility	7.97 (3.42)	7.40 (3.27)	8.54 (3.48)	0.34
Eccentricity	7.11 (3.39)	6.89 (3.28)	7.34 (3.48)	0.13
Emotional Lability	9.21 (3.14)	8.28 (2.85)	10.13 (3.15)	0.62
Grandiosity	5.50 (2.12)	5.74 (2.32)	5.25 (1.88)	0.23
Hostility	6.95 (2.85)	6.35 (2.69)	7.54 (2.88)	0.43
Impulsivity	6.47 (2.91)	6.10 (2.79)	6.85 (3.02)	0.26
Intimacy Avoidance	6.26 (3.26)	5.99 (2.76)	6.54 (3.66)	0.17
Irresponsibility	5.51 (1.91)	5.52 (2.01)	5.50 (1.80)	0.01
Manipulativeness	5.62 (2.13)	5.72 (2.18)	5.51 (2.08)	0.10
Perceptual Dysregulation	5.32 (2.20)	5.21 (2.13)	5.44 (2.25)	0.10
Perseveration	7.31 (2.96)	6.88 (2.79)	7.75 (3.07)	0.30
Restricted Affectivity	7.35 (2.66)	7.67 (2.68)	7.03 (2.59)	0.24
Rigid Perfectionism	8.06 (3.12)	7.84 (3.05)	8.27 (3.19)	0.14
Risk Taking	5.40 (2.23)	5.51 (2.30)	5.29 (2.14)	0.10
Separation Insecurity	7.90 (3.26)	7.78 (3.26)	8.03 (3.26)	0.08
Submissiveness	6.94 (2.82)	6.63 (2.56)	7.25 (3.02)	0.22
Suspiciousness	6.69 (2.57)	6.50 (2.51)	6.88 (2.63)	0.15
Unusual Beliefs and Exp.	5.92 (2.67)	5.63 (2.45)	6.21 (2.84)	0.22
Withdrawal	6.98 (2.93)	6.86 (2.86)	7.09 (3.00)	0.08
Alcohol-Use Disorder criteria				
Larger/Longer	0.17 (0.17)	0.20 (0.40)	0.14 (0.35)	0.16
Quit/Control	0.19 (0.23)	0.25 (0.43)	0.13 (0.33)	0.31
Time spent	0.05 (0.20)	0.06 (0.24)	0.04 (0.20)	0.09
Activities given up	0.04 (0.38)	0.05 (0.22)	0.03 (0.17)	0.10
Tolerance	0.05 (0.39)	0.07 (0.25)	0.03 (0.18)	0.18
Withdrawal	0.05 (0.22)	0.06 (0.24)	0.04 (0.20)	0.09
Phys/Psych.Problems	0.08 (0.20)	0.09 (0.28)	0.06 (0.24)	0.12
Neglect roles	0.03 (0.22)	0.04 (0.21)	0.02 (0.13)	0.11
Hazardous use	0.06 (0.22)	0.08 (0.27)	0.03 (0.18)	0.21
Social/Interp.Problems	0.04 (0.26)	0.05 (0.22)	0.03 (0.18)	0.10
Craving	0.06 (0.24)	0.07 (0.26)	0.05 (0.22)	0.08

**Table 5 jcm-11-03468-t005:** Partial and zero-order correlations between network symptoms on estimation sample.

Scales	1	2	3	4	5	6	7	8	9	10	11	12	13	14	15	16	17	18
1. Callousness	1	0.45	0.31	0.29	0.49	0.25	0.36	**0.14**	**0.13**	**0.25**	**0.27**	**0.31**	**0.32**	**0.26**	**0.31**	**0.29**	**0.33**	**0.31**
2. Grandiosity	0.12	1	0.29	0.29	0.36	0.39	0.43	**0.13**	**0.12**	**0.20**	**0.21**	**0.29**	**0.23**	**0.19**	**0.20**	**0.26**	**0.26**	**0.29**
3. Hostility	0.01	-	1	0.57	0.42	0.41	0.38	**0.13**	**0.08**	**0.18**	**0.20**	**0.18**	**0.23**	**0.21**	**0.19**	**0.19**	**0.21**	**0.19**
4. Impulsivity	-	-	0.20	1	0.53	0.31	0.53	**0.13**	**0.08**	**0.21**	**0.25**	**0.19**	**0.23**	**0.23**	**0.20**	**0.20**	**0.21**	**0.17**
5. Irresponsibility	0.12	-	0.00	0.16	1	0.23	0.43	**0.23**	**0.13**	**0.28**	**0.35**	**0.29**	**0.31**	**0.33**	**0.34**	**0.35**	**0.32**	**0.34**
6. Rigid Perfectionism	-	0.12	0.04	0.13	−0.02	1	0.33	**0.04**	**0.06**	**0.08**	**0.08**	**0.07**	**0.07**	**0.10**	**0.05**	**0.13**	**0.13**	**0.13**
7. Risk taking	0.03	0.12	-	0.18	-	-	1	**0.25**	**0.18**	**0.27**	**0.28**	**0.30**	**0.31**	**0.35**	**0.29**	**0.35**	**0.30**	**0.30**
8. Larger/longer	**-**	**-**	**-**	**-**	**-**	**-**	**0.05**	1	0.48	0.35	0.28	0.39	0.30	0.42	0.34	0.36	0.34	0.34
9. Quit/control	**-**	**-**	**-**	**-**	**-**	**-**	**-**	0.27	1	0.39	0.33	0.32	0.29	0.40	0.32	0.29	0.29	0.25
10. Time Spent	**-**	**-**	**-**	**-**	**-**	**-**	**0.01**	0.19	0.23	1	0.58	0.52	0.48	0.55	0.48	0.46	0.50	0.40
11. Activities given up	**-**	**-**	**-**	**-**	**0.04**	**-**	**-**	-	-	0.20	1	0.49	0.50	0.50	0.53	0.52	0.51	0.48
12. Tolerance	**-**	**-**	**-**	**-**	**-**	**-**	**0.02**	0.04	-	0.22	0.09	1	0.51	0.52	0.51	0.44	0.53	0.48
13. Withdrawal	**0.03**	**-**	**0.01**	**-**	**0.02**	**-**	**0.04**	0.08	-	0	0.10	0.19	1	0.55	0.49	0.45	0.46	0.41
14. Phys./Psych.Problems	**-**	**-**	**-**	**-**	**-**	**-**	**0.01**	0.08	0.05	0.12	0.05	0.05	0.20	1	0.57	0.57	0.54	0.48
15. Neglect roles	**0.05**	**-**	**-**	**-**	**0.01**	**0.02**	**-**	-	-	0.02	0.22	0.11	0.11	0.09	1	0.54	0.66	0.42
16. Hazardous use	**0.01**	**-**	**-**	**-**	**-**	**-**	**0.04**	0.10	-	0.04	0.05	0.02	-	0.17	0.13	1	0.59	0.48
17. Social/Interp.Problems	**-**	**0.02**	**-**	**-**	**-**	**-**	**0.01**	-	-	0.05	0.06	0.10	0.09	0.11	0.25	0.21	1	0.42
18. Craving	**0.03**	**0.02**	**-**	**-**	**0.05**	**-**	**-**	0.10	0.05	-	-	0.27	0.08	0.11	0.01	0.06	0.04	1

Partial correlations are shown on the lower diagonal and zero-order correlations on the upper diagonal. The dashes represent correlation values = 0. The bold type reflects the relationships between the bridging nodes and the AUD criteria.

## Data Availability

Database should be requested to the correspondence author.

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
