# Peer review of "Bridge Nodes between Personality Traits and Alcohol-Use Disorder Criteria: The Relevance of Externalizing Traits of Risk Taking, Callousness, and Irresponsibility"

_jcm, 2022, doi:10.3390/jcm11123468_

Round 1

Reviewer 1 Report

The paper focuses on an interesting objective:  explore the existing comorbidity between personality disorders and alcohol use disorder.

The theme is well founded through a consistent literature review. The objectives are correctly defined. The methodology is well-designed and is consistent with the objectives of the study. The interpretation and discussion of results is clear, objective, and consistent. The discussion summarize well the results obtained and are consistent with the work presented. Only a few minor details remain:

1)    p.1 (line 11) - lack of space in “742were”. Also, it would be better to put a colon after "participants" so as not to start the next sentence with a number

2)      p.7 (line 245) - I suggest adding the subtitles for PC and PR

3)      p.8 - It was important to better clarify the appearance of the“Bridging nodes” between AUD criteria and personality facets from a statistical point of view.

4)      p.10 (lines 317-318) - put "Irresponsibility" in lower case like the other mentioned facets

5)      A clearly separate section of the "Discussion" entitled "Final Considerations" or "Conclusions" should be created which summarizes the main conclusions of the investigation

  I really enjoyed reading your work! It is very clear and well organized! Congratulations!

Author Response

The authors would like to thank you for your comments. We will now proceed to respond to the suggestions you have made.

1)    p.1 (line 11) - lack of space in “742were”. Also, it would be better to put a colon after "participants" so as not to start the next sentence with a number

  We appreciate the detection of the typo and your suggestion. We have now introduced these changes.   

2)      p.7 (line 245) - I suggest adding the subtitles for PC and PR

We have now indicated the meaning of the acronyms PC and PR in a footnote in Table 2.  

3)      p.8 - It was important to better clarify the appearance of the “Bridging nodes” between AUD criteria and personality facets from a statistical point of view.

 We welcome your suggestions, as it may indeed be useful for readers to clarify these concepts. To this end, in the "data analysis" section we have proceeded to further explain the concepts associated with bridge nodes.  In addition, in the results section we have proceeded to explain the results associated with the bridge nodes and we have also included the values in Table 3 (bridge strength, bridge EI1, and bridge EI2). 

On the other hand, the authors noticed an error in the MS, since line 319 referred to a table with the descriptive statistics of the AMPD facets and the AUD criteria in the total sample, (the sample of men and women) but this did not appear in the MS. We have now included this table.    

4)      p.10 (lines 317-318) - put "Irresponsibility" in lower case like the other mentioned facets

 The authors have proceeded to modify the text as indicated

 5)      A clearly separate section of the "Discussion" entitled "Final Considerations" or "Conclusions" should be created which summarizes the main conclusions of the investigation

Following the reviewer's suggestions, we have proceeded to introduce some final conclusions.

I really enjoyed reading your work! It is very clear and well organized! Congratulations!

Thank you very much

Please see the attached file to view the MS with the changes noted above

Reviewer 2 Report

Thank you for sending the manuscript. The importance of the topic is accepted, by and large, the clinical importance of the finding is limited. My additional comments are listed below:

1. The title should be rewritten to be more clear.

2. The conclusion of the abstract is not relevant enough to the results of the abstract.

3. Inclusion criteria and methods should be explained in more details in the abstract.

4. Cultural aspect of the topic should be emphasized in both background and discussion.

5. Age range of the participants was repeated three times in the method.

6. Detailed demographic data should be summarized in a table and removed from the text.

7. Measures should be explained in more details.

Author Response

The authors would like to thank you for your comments. We will now proceed to respond to the suggestions you have made.

Comments and Suggestions for Authors

Thank you for sending the manuscript. The importance of the topic is accepted, by and large, the clinical importance of the finding is limited. My additional comments are listed below:

  1. The title should be rewritten to be more clear.

We thank the reviewer for this comment. The authors considered the following title to be more explanatory: Bridge nodes between personality traits and alcohol use disorder criteria: the relevance of externalizing traits of risk taking, callousness, and irresponsibility

  1. The conclusion of the abstract is not relevant enough to the results of the abstract.

Following the reviewer's suggestions, we have changed the conclusions of the abstract.

  1. Inclusion criteria and methods should be explained in more details in the abstract.

We thank the reviewer for this suggestion. However, given the word limitation for the abstract (200 words), we have tried to provide more clarification on these suggestions. We hope this is satisfactory, and if not, please let us know. Thank you

  1. Cultural aspect of the topic should be emphasized in both background and discussion.

We are very grateful for your comment because, as you point out, it is essential to contextualize the results culturally. Following your indications, and to reflect the cultural aspects of the results found, we have included two paragraphs in the introduction and in the discussion. We believe that through the text introduced, readers will be aware of the need to contextualize the findings in countries and cultures similar to those in which the study has been conducted.

  1. Age range of the participants was repeated three times in the method.

We appreciate your comment. In the inclusion criteria we have replaced "being between 18 and 80" years old with "being over 18 years old" so that "between 18 and 80" is not repeated three times.

  1. Detailed demographic data should be summarized in a table and removed from the text.

Following these suggestions, we have created a new table with the sociodemographic characteristics (Table 1) and have modified the order of the rest of the tables acordingly.

  1. Measures should be explained in more details.

We now indicate which facets are assessed by the PID-5-SF. We have also considered it relevant to specify that the time frame for evaluation of the AUD criteria with the SDSS was 30 days prior to the interview.

Thank you very much for your review. Please see the attached file to view the MS with the changes noted above

Round 2

Reviewer 2 Report

Thank you for sending the revised manuscript. I still believe that the clinical importance of the manuscript is limited and it prevents me to suggest it for publication in the journal.